# Dietary Iodine Intake and Related Factors among Secondary School Students in Macao

**DOI:** 10.3390/healthcare11101472

**Published:** 2023-05-18

**Authors:** Cleandy Lei, Xiaoyu Tao, Ying Xiao

**Affiliations:** Faculty of Medicine, Macau University of Science and Technology, Taipa, Macao SAR 999078, China; cleandy_lei@yahoo.com.tw (C.L.); derektao@zcst.edu.cn (X.T.)

**Keywords:** iodine, Macao secondary school students, food frequency questionnaire, median daily iodine intake, urinary iodine

## Abstract

Iodine is a crucial micronutrient that is indispensable for optimal physical growth and cognitive maturation. However, the dietary iodine intake status of Macao’s population is unknown. Therefore, a cross-sectional study was conducted to assess the dietary iodine intake of Macao secondary school students. Four hundred and twenty-four students filled in a self-developed, 61-item, iodine-specific food frequency questionnaire (I-FFQ). The dietary iodine intake was calculated based on the I-FFQ and food composition database. The median daily iodine intake of the students was 74.4 µg, which is lower than the 150 µg recommended by the World Health Organization (WHO). The intake frequency of dried seaweed and kelp was also low, with 49.3% and 64.2% of students consuming these foods infrequently over a month. In conclusion, the dietary iodine intake of secondary school students in Macao was inadequate. It is recommended that individuals should take the initiative to gain iodine-related knowledge. Students are advised to eat a variety of iodine-rich foods, such as seaweed and seafood, as part of a healthy, balanced diet to ensure sufficient iodine intake. Furthermore, it will be necessary to measure urinary iodine for the iodine status assessment of the Macao population in future.

## 1. Introduction

Iodine is an essential micronutrient for the human body. It is an indispensable component for the synthesis of thyroid hormones and plays important roles in promoting body growth and brain development, regulating metabolism, and maintaining normal function of the cardiovascular and digestive systems. Either excessive or insufficient iodine intake will induce abnormal functioning of the thyroid gland, thereby causing adverse effects on the body. The Wolff–Chaikoff effect, an acute and temporary inhibition of the synthesis and secretion of thyroid hormones, occurs in cases of excessive iodine intake. Normally, the body will adapt to and escape from this effect after a few weeks due to the auto-regulatory mechanism of the thyroid gland. However, the long-term intake of excessive iodine will lead to the imbalance and dysfunction of this self-regulation, resulting in hyperthyroidism or hypothyroidism, as well as autoimmune thyroiditis. Iodine deficiency can cause varying degrees of damage to the body depending on various factors, including the severity of deficiency, time of occurrence in the life cycle and differences in individual responses to iodine deficiency. A compensatory goiter will be induced even when one is mildly deficient in iodine. Other health consequences of iodine deficiency include stillbirth and congenital anomalies in the fetus, cretinism and increased infant mortality among neonates, impaired mental function and delayed physical development in children and adolescents, iodine-induced hyperthyroidism in adults and spontaneous abortion in pregnant women. These symptoms are collectively known as iodine deficiency disorders (IDD) [1,2].

Approximately 80% of the iodine required for the human body can be obtained through dietary intake, while drinking water accounts for 10–20% and atmospheric sources provide less than 5% [2]. Marine products from the ocean are the main sources of iodine in nature, such as seaweed, kelp, shellfish and seawater fish [3]. Even if the intake of iodine in the diet is insufficient, the iodine metabolism cycle will accelerate and maintain the normal function of the body in a compensatory form. When iodine intake is completely absent, the iodine store in the body still can meet its needs for the next 2–3 months. However, IDD will be triggered when compensatory mechanisms are insufficient to cope with the body’s needs [2].

IDD can significantly restrain intellectual development and therefore represents a leading global nutritional challenge that poses a threat of brain damage. Studies have shown that children residing in regions with inadequate iodine intake exhibited an ap-proximate 13.5-point reduction in intelligence quotient (IQ) scores. This reflects that iodine deficiency affects the learning ability of school-age children and even their quality of life and long-term productivity in those regions [1]. A longitudinal study of parents and children from the United Kingdom assessed the association between maternal iodine status and children’s IQ at the age of 8 years and reading ability at the age of 9 years. The data supported the hypothesis that low maternal iodine status was associated with an increased risk of sub-optimal scores for IQ at age 8 and reading accuracy, in addition to comprehension and reading scores, at age 9 [4]. Shenzhen of Guangdong Province, China, was originally a mildly iodine-deficient area. Since the implementation of salt iodization measures in 1996, the average IQ score of 8–10-year-old school-age children has shown an upward trend, increasing from 100.88 in 2002 to 108.45 in 2014 [5].

In accordance with statistical data provided by the World Health Organization (WHO), one can extrapolate whether a country’s population is afflicted by iodine deficiency from the prevalence of iodine insufficiency among pupils aged 6–12 years. The current estimates suggest that the proportion of individuals affected by inadequate iodine intake globally is as high as 31%, with particularly concentrated rates observed within regions of Southeast Asia and Europe [1]. Although the condition of IDD may vary in severity across different countries, it can be prevented conveniently and economically. Compared with other micronutrient deficiencies, it is less affected by low income or poor-quality in diets of the population [6]. The WHO pointed out that the serious health effects caused by a lack of iodine could be eliminated by adding just a small amount of iodine to table salt in daily life; thus, governments around the world should actively face and address this public health problem [1].

Considering that adolescents are in a stage of growth and development and have an increased demand for iodine and are also more autonomous in their dietary choices and begin to establish dietary patterns in adulthood, this study aimed to evaluate iodine intake, the main food sources of iodine and the intake frequency of iodine-rich foods among secondary school students in Macao, build a foundation for iodine-related nutrition research and draw public attention to adequate iodine intake in Macao.

## 2. Materials and Methods

### 2.1. Study Design and Recruitment

A cross-sectional descriptive research method was used. A total of 27,627 Macao secondary school students in the 2020/2021 academic year were enrolled as the research population [7]. Stratified random cluster sampling was conducted in 3 schools selected from 47 schools. One class was randomly selected from every level of each school, resulting in eighteen classes being obtained as samples.

### 2.2. Procedure

Data were collected from 18 classes among 3 schools from December 2020 to February 2021. After obtaining agreement and contacting the directors of the secondary schools, questionnaires were distributed to each class. A slideshow of food pictures with the names of the food items, portion sizes and actual tableware were shown by the researcher on-site. The students filled in the forms by themselves based on estimations of the serving consumed.

### 2.3. Ethical Aspects

Participation was voluntary, and the questionnaire was anonymous. In addition to being included in the questionnaire, the research objectives were also explained to the research subjects before the survey was administered. During the process, they had the right to refuse participation and withdraw from the research at any time. The questionnaires and materials obtained from the research survey were kept strictly confidential. This study was approved by the Faculty of Medicine of the Macau University of Science and Technology.

### 2.4. Iodine-Specific Food Frequency Questionnaire (I-FFQ)

The questionnaire was designed based on the research objectives and literature, and then its reliability and validity were verified by 7 experts in the areas of public health, nutrition and epidemiology. From the data obtained through the survey, as well as the question on iodized salt use status at home, we could understand the frequency and amounts of iodine-rich foods consumed by the subjects.

An I-FFQ developed in Australia was utilized as a reference to design the self-developed I-FFQ [8]. Data from a survey of food iodine content in Hong Kong and food composition tables of China and Japan were used for the selection of food items and analysis of iodine content [9,10,11]. As shown in Table 1, there were 13 food groups with a total of 61 food items in the self-developed I-FFQ, including (i) cereals and grains, (ii) starchy rhizomes, (iii) soybean products, (iv) vegetables and legumes, (v) seaweed and mushrooms, (vi) fruits, (vii) nuts and seeds, (viii) livestock, (ix) poultry, (x) milk and dairy products, (xi) eggs, (xii) seafood and (xiii) other common foods and drinks. Pictures of food that showed food items and portion sizes were used as a supplementary tool to help the subjects to distinguish between various types of foods and their amounts.

### 2.5. Data Analysis

The iodine intake was derived from the food composition database of China and Japan according to the weight or volume of each food item [10,11]. Descriptive, inferential and regression statistics were determined using IBM SPSS 24.0 (IBM, Armonk, New York, USA). Linear regression and logistic regression were used to assess the associations between sociodemographic characteristics and iodine intake and the associations between sociodemographic characteristics and iodized salt use status, respectively. One-way ANOVA was conducted to determine the association between iodized salt use status and daily iodine intake. Statistical significance was set at *p* < 0.05.

## 3. Results

### 3.1. Sociodemographic Characteristics

A total of 432 questionnaires were collected. After screening and eliminating the questionnaires that were missed or filled-in inappropriately, a total of 424 valid questionnaires were obtained, with an effective rate of 98.1%. Table 2 shows the distribution of the subjects according to various sociodemographic characteristics.

### 3.2. Iodine Intake from Various Food Groups

The median iodine intakes of the subjects from various food groups were estimated and are shown in Table 3. The results indicated that the median daily iodine intake of the subjects was only 74.4 µg, which is less than half of the 150 µg recommended by the WHO. The largest contributor to iodine intake was milk and dairy products, with a median of 10.4 µg. This was followed by other common foods and drinks at 8.9 µg, seafood at 8.3 µg, eggs at 7.6 µg, seaweed and mushrooms at 5.8 µg, vegetables and legumes at 4.0 µg, cereals and grains at 1.8 µg, livestock at 1.4 µg and soybean products at 1.3 µg. The median iodine intakes from the groups of fruits, poultry, starchy rhizomes and nuts and seeds were less than 1.0 µg, respectively. The median iodine intake of each food item among the subjects is shown in Appendix A.

### 3.3. Intake Frequency of Iodine-Rich Foods

The food items with higher iodine (>10 µg/100 g) in the I-FFQ used in this study were selected for an analysis of their iodine contribution and the intake frequency of the subjects (Table 4). The percentages of subjects who consumed the most iodine-rich foods daily, including dried seaweed, seaweed snacks, kelp soup, shellfish and kelp, were all less than 5%, respectively. The percentages of subjects who rarely ate these five food items in the past month were as high as 49.3%, 53.8%, 82.1%, 44.6% and 64.2%, respectively.

### 3.4. Associations between Sociodemographic Characteristics and Iodine Intake Status

Table 5 shows a significant negative association between the grades of the subjects and their daily iodine intake (*p* < 0.05), indicating that the higher the grade of a subject was, the lower their daily iodine intake was. However, there was no significant difference in daily iodine intake between the variables of gender, age and parents’ educational levels.

### 3.5. Associations between Sociodemographic Characteristics and Iodized Salt Consumption Status at Home

Table 6 shows that the types of iodized salt used in the subjects’ homes were affected by their gender. In total, 15.5% of boys answered that iodized salt was used in their homes, while this was the case for only 7.4% of girls. There was no significant difference in the type of salt used at home in terms of the students’ age and grade and the parents’ educational levels. However, among all the subjects, more than 70% of them did not know the type of salt used in their homes, with a significantly higher percentage among girls (85.6%) than among boys (73.4%) (*p* < 0.01).

### 3.6. Association between Iodized Salt Consumption Status at Home and Daily Iodine Intake

In Table 7, the iodine intake of the subjects was statistically different depending on the type of salt in the home. After comparison, the iodine intake of the study subjects who had used iodized salt was found to be significantly higher (*p* < 0.001).

## 4. Discussion

Currently, mandatory salt iodization is legislatively mandated in over 120 countries, whereas voluntary salt iodization is practiced in more than 20 countries [12]. Overall, the percentage of the world’s population who consume iodized salt is 89% [13]. The 2021 annual report of the Iodine Global Network (IGN) evaluated the general populations of 111 countries, which adequate iodine intake exhibited in a total of 141 nations with available data. This represents a two-fold increase from the levels observed over the past two decades, which highlights the efficacy of salt-iodization-based interventions in combating IDD. However, despite this positive trend, there remain 19 countries in which inadequate iodine intake persists owing to factors such as the low coverage and suboptimal utilization of iodized salt [14].

The major methods of iodine supplementation in the diet are using iodized salt and eating iodine-rich foods. In addition, nutritional supplements can also be used as a source of iodine. Based on the WHO recommendation, the daily iodine intake for adolescents is 150 µg. However, from the results of the I-FFQ in this study, it was found that the median daily iodine intake of secondary school students in Macao was only 74.4 µg, which is an inadequate level, and this also tended to decrease with age or grade. Although the daily iodine intake was significantly higher among iodized salt users, overall, nearly 90% of the subjects did not use iodized salt and were not aware of the type of salt they used at home. Apart from the use of iodized salt, the subjects did not consume seaweed, mushrooms and seafood as the main sources of iodine in their diets and did not use these types of foods with higher iodine to compensate for their insufficient iodine intake due to a lack of use of iodized salt. Thus, without the habits of using iodized salt or obtaining enough iodine from the diet and increasing the frequency of iodine intake, the risk of iodine deficiency will be greatly increased.

As early as 1994, the WHO and United Nations International Children’s Education Fund (UNICEF) recommended Universal Salt Iodization (USI) as a safe, economic and sustainable strategy to ensure adequate iodine intake for populations at a health policy joint meeting. Nearly all countries with iodine deficiency agree that USI is the most cost- effective countermeasure for eliminating IDD. The USI covers the addition of iodine to salt used for humans and livestock, including salt used in the food industry. The sufficient iodine supplementation of table salt provides adequate iodine availability to the population, and this needs to be implemented on a continuous and self-sustaining basis. The WHO recommendation for iodine added to table salt is 20–40 mg/kg. Countries have also formulated different recommendations according to the needs of their residents [1], such as Australia, with 25–65 mg/kg, New Zealand, with 25–65 mg/kg [15], France, with 18 mg/kg, Germany, with 15–25 mg/kg, Denmark, with 20 mg/kg [16], and Switzerland, with 20–25 mg/kg [17]. The “Iodine of edible salt” (GB 26878-2011) standard issued by China in 2012 stipulates that the average amount of iodine added to edible salt should be 20–30 mg/kg, and the allowable fluctuation range is ±30% [18].

The prevention and treatment of iodine deficiency disorders is a public health issue of global concern. Many countries have actively performed the evaluation and monitoring of people’s iodine intake and formulated strategies and regulations according to local conditions to improve the iodine intake of their residents. On the contrary, there are currently no iodine-related supplementation measures, iodine intake assessments or research data in Macao, and the use of iodized salt in Macao is not widespread. Therefore, continuous assessment and the regular adjustment of measures are required to ensure that each population maintains an appropriate level of iodine intake in different life stages.

The median urinary iodine is conventionally used as an objective biological indicator to evaluate the iodine nutritional status of the population. However, due to various considerations such as human resources, time limitations and respondents’ acceptance, the median urinary iodine indicator is more likely to be affected by recent iodine intake. Thus, a food frequency questionnaire was used to evaluate the long-term dietary iodine intake of the subjects in this study.

The advantage of using the FFQ as a dietary survey tool is that it can be used to quickly obtain the types and amounts of foods that are often consumed, which can reflect the long-term nutrient intake pattern and serve as a basis for studying the relationship between dietary patterns and chronic diseases. The results can also be used as a reference for dietary guidance and the health education of the population. Countries currently using the FFQ to assess iodine nutritional status include the United States, Finland, Colombia [19], Australia [8], New Zealand [20], the UK [21], Poland [22], Denmark [23], Norway [24], China [25] and Malaysia [26]. With the popularization of the internet, many countries have carried out online dietary evaluations through the internet and mobile phone applications [27,28]. Compared with paper questionnaires, the FFQ is not only more accurate, efficient and cost-effective but also reduces errors caused by human operations and increases acceptance among participants [29]. For example, food frequency questionnaire survey methods include “Food4Me” in Europe [30], the “SFFFQ” in the UK [31] and the “QFFQ” in South Africa [32]. The 24 h diet review methods include the “ASA24” in the United States [33], “Oxford WebQ” in the United Kingdom [34], “myfood24” [35] and “ INTAKE24” [36].

Although the reliability of the iodine food frequency questionnaire has been confirmed by many studies, the questionnaire has been widely used [8,21,25], and many countries have carried out online dietary evaluations on the internet and mobile applications [27], the food frequency questionnaire method used here can only analyze the iodine intake of the research subjects from a dietary perspective. There are still shortcomings that may lead to errors, including the facts that the survey method is qualitative rather than quantitative, cooking oils and seasonings can easily be overlooked, and food portion estimation can be inaccurate. Hence, it is necessary to perform median urinary iodine measurements for Macao residents, as this is the most accurate and objective indicator for evaluating the iodine intake status of the population. From the results of the median urinary iodine measurements, we can formulate comprehensive regulations and action plans for achieving the goal of IDD prevention.

Moreover, individuals should proactively seek to gain more iodine-related knowledge and understand the recommended intake of iodine for their age. They should also use iodized salt in their daily diets according to their dietary habits and moderately consume iodine-rich foods such as seaweed, kelp, shellfish and seawater fish to ensure that their iodine intake is within the appropriate range. Irrespective of whether iodine is sourced through iodized salt or natural food, it is essential to account for the potential loss of iodine or other essential nutrients due to various environmental conditions, including cooking and storage. A judicious approach to cooking techniques can minimize nutrient loss by reducing the duration of exposure to high temperatures. Cooking methods such as steaming and stir-frying are preferable over high-temperature techniques such as deep-frying, which may lead to a diminished iodine content when preparing dishes. Moreover, iodized salt should be added to meals after culinary preparation but prior to serving in order to maximize the preservation of the iodine content. Finally, the storage of iodized salt in an opaque, tightly sealed container has been found to be effective in maintaining its iodine content while protecting it from light [9].

## 5. Conclusions

The dietary iodine intake of secondary school students in Macao did not meet the WHO recommendation. Their main sources of iodine were not iodized salt or iodine-rich foods. It is suggested that students should learn more about the health benefits of iodine and their own recommended iodine intake. They should also use iodized salt as a replacement for common table salt and moderately consume iodine-rich foods such as seaweed and kelp in order to achieve “scientific iodine supplementation” and maintain normal body growth and development. Furthermore, excessive or insufficient iodine intake can only be determined through biochemical tests, as the thyroid-related symptoms of both conditions are similar. We recommend measuring urinary iodine among Macao residents in the future in order to assess the iodine nutritional status more accurately for each population. Additionally, we should formulate corresponding measures and promote nutritional education to ensure that each population has a sufficient iodine intake maintained at the appropriate level.

## Figures and Tables

**Table 1 healthcare-11-01472-t001:** Thirteen food groups in the I-FFQ.

Food Group	Food Quantity
Cereals and grains	6
Starchy rhizomes	1
Soybean products	3
Vegetables and legumes	6
Seaweed and mushrooms	3
Fruits	9
Nuts and seeds	3
Livestock	4
Poultry	2
Milk and dairy products	3
Eggs	2
Seafood	8
Other common foods and drinks	11
Total	61

**Table 2 healthcare-11-01472-t002:** Sociodemographic characteristics of the subjects (*n* = 424).

Variable	Sub-Variable	*n*	%
Gender	Male	154	36.3
	Female	270	63.7
Age (years)	≦17	298	70.3
	≧18	126	29.7
Grade	Junior 1	77	18.2
	Junior 2	75	17.7
	Junior 3	80	18.9
	Senior 1	77	18.2
	Senior 2	59	13.9
	Senior 3	56	13.2
Father’s	Primary level or below	46	10.8
educational level	Lower secondary level	64	15.1
	Upper secondary level	86	20.3
	Bachelor’s degree or tertiary education	90	21.2
	Master’s degree or above	16	3.8
	Unknown	122	28.8
Mother’s	Primary level or below	29	6.8
educational level	Lower secondary level	79	18.6
	Upper secondary level	86	20.3
	Bachelor’s degree or tertiary education	99	23.3
	Master’s degree or above	24	5.7
	Unknown	107	25.2

**Table 3 healthcare-11-01472-t003:** Iodine intakes of the subjects from various food groups.

Food Group	Mean ± SD (µg)	Median (µg)
Milk and dairy products	16.0 ± 18.1	10.4
Other common foods and drinks	22.9 ± 52.1	8.9
Seafood	14.1 ± 24.5	8.3
Eggs	12.3 ± 13.1	7.6
Seaweed and mushrooms	15.7 ± 41.5	5.8
Vegetables and legumes	4.8 ± 4.0	4.0
Cereals and grains	2.1 ± 1.4	1.8
Livestock	3.0 ± 4.6	1.4
Soybean products	2.8 ± 5.2	1.3
Fruits	1.5 ± 2.1	0.8
Poultry	0.4 ± 1.0	0.2
Starchy rhizomes	0.2 ± 0.4	0.2
Nuts and seeds	0.3 ± 0.9	0.0
Daily iodine intake	96.1 ± 97.2	74.4

**Table 4 healthcare-11-01472-t004:** Main food sources of iodine and their intake frequencies (*n* = 424).

Food Item	Iodine/100 g (µg)	Per Serving		Proportion of Subjects Who Ate at Least Half a Serving (%)			Rarely in the Past Month
		Iodine (µg)	Weight (g)	Per Day	Per Week	Per Month	
Dried seaweed	6600.0	66.0	1.0	3.1	11.8	35.8	49.3
Seaweed snack	3400	34.0	1.0	4.2	11.6	30.4	53.8
Kelp soup	363.4	726.8	200.0	0.2	1.7	16.0	82.1
Shellfish	140.0	21.0	15.0	1.2	11.8	42.4	44.6
Kelp	113.9	113.9	100.0	1.2	25.2	9.4	64.2
Crab	47.0	56.4	120.0	1.4	7.3	35.4	55.9
Fish ball	30.0	15.0	50.0	3.5	25.0	38.9	32.6
Yogurt	29.0	34.8	120.0	4.5	17.0	26.4	52.1
Egg	29.0	17.4	60.0	33.5	45.0	16.5	5.0
Cheese	25.0	5.0	20.0	6.4	26.7	31.1	35.8
Preserved egg	23.0	16.1	70.0	1.4	5.7	24.5	68.4
Seawater fish	17.0	17.0	100.0	7.8	27.8	21.9	42.5
Shrimp	16.1	8.1	50.0	3.3	20.7	43.9	32.1
Sausage and ham	16.0	8.0	50.0	10.1	35.4	34.4	20.1
Cake	14.8	11.8	80.0	2.5	23.1	48.0	26.4
Egg waffle	14.2	14.2	100.0	1.6	12.3	43.9	42.2
Mushroom	13.5	13.5	100.0	5.4	36.6	33.7	24.3
Pistachio	10.3	5.2	50.0	0.5	2.4	15.3	81.8

**Table 5 healthcare-11-01472-t005:** Associations between sociodemographic characteristics and iodine intake status.

Variable	Daily Iodine Intake
Gender	−0.145
Age	−0.041
Grade	−0.086 *
Father’s education level	0.046
Mother’s education level	−0.060

* *p* < 0.05.

**Table 6 healthcare-11-01472-t006:** Associations between sociodemographic characteristics and iodized salt consumption status at home (*n* (%)).

Variable	Sub-Variable	Iodized Salt	Mixed Use of Iodized Salt and Non-Iodized Salt	Non-Iodized Salt	Unknown	*n*	*p*-Value
Gender	Male	19 (12.3)	5 (3.2)	17 (11.0)	113 (73.4)	154	<0.01 **
	Female	6 (2.2)	14 (5.2)	19 (7.0)	231 (85.6)	270	
Age (years)	≦17	18 (6.0)	14 (4.7)	24 (8.1)	242 (81.2)	298	0.869947
	≧18	7 (5.6)	5 (4.0)	12 (9.5)	102 (81.0)	126	
Grade	Junior 1	7 (9.1)	3 (3.9)	3 (3.9)	64 (83.1)	77	0.792
	Junior 2	3 (4.0)	4 (5.3)	7 (9.3)	61 (81.3)	75	
	Junior 3	6 (7.5)	2 (2.5)	11 (13.8)	61 (76.3)	80	
	Senior 1	5 (6.5)	5 (6.5)	6 (7.8)	6 (79.2)	77	
	Senior 2	4 (6.8)	3 (5.1)	5 (8.5)	47 (79.7)	59	
	Senior 3	0 (0.0)	2 (3.6)	4 (7.1)	50 (89.3)	56	
Father’s educational level	Primary level or below	4 (8.7)	3 (6.5)	4 (8.7)	35 (76.1)	46	0.566
	Lower secondary level	5 (7.8)	4 (6.3)	6 (9.4)	49 (76.6)	64	
	Upper secondary level	4 (4.7)	4 (4.7)	7 (8.1)	71 (82.6)	86	
	Bachelor’s degree	6 (6.7)	5 (5.6)	11 (12.2)	68 (75.6)	90	
	Master’s degree or above	0 (0.0)	0 (0.0)	2 (12.5)	14 (87.5)	16	
	Unknown	6 (4.9)	3 (2.5)	6 (4.9)	107 (87.7)	122	
Mother’s educational level	Primary level or below	0 (0.0)	1 (3.4)	2 (6.9)	26 (89.7)	29	0.411
	Lower secondary level	8 (10.1)	6 (7.6)	9 (11.4)	56 (70.9)	79	
	Upper secondary level	6 (7.0)	3 (3.5)	8 (9.3)	69 (80.2)	86	
	Bachelor’s degree	5 (5.1)	5 (5.1)	9 (9.1)	80 (80.8)	99	
	Master’s degree or above	0 (0.0)	1 (4.2)	5 (20.8)	18 (75.0)	24	
	Unknown	6 (5.6)	3 (2.8)	3 (2.8)	95 (88.8)	107	

** *p* < 0.01.

**Table 7 healthcare-11-01472-t007:** Association between iodized salt consumption status at home and daily iodine intake.

Type of Salt Used at Home	*n* (%)	Daily Iodine Intake (µg)		*p*-Value
		Mean ± SD	Median	
Iodized salt	25 (5.9)	149.0 ± 132.8	113.9	<0.001 ***
Mixed use of iodized salt and non-iodized salt	19 (4.5)	165.1 ± 176.3	119.3	
Non-iodized salt	36 (8.5)	127.7 ± 130.1	84.2	
Unknown	344 (81.1)	85.1 ± 79.8	69.0	

*** *p* < 0.001.

## Data Availability

The data presented in this study are available from the corresponding author upon reasonable request, while the data collected from the survey are not publicly available due to confidentiality.

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
