# Peer review of "Dietary Iodine Intake and Related Factors among Secondary School Students in Macao"

_healthcare, 2023, doi:10.3390/healthcare11101472_

Round 1

Reviewer 1 Report

Major concerns:

1. The efficacy of FFQ is directly linked to the ability of the subject to understand the question and to answer correctly. Given the large part of underage subjects, authors should better state how they made the questionnaire suitable for this audience

2. The main source of iodine is by the use of iodized salt, as correctly stated by the authors. Given that the vast majority of the subjects did not know the kind of salt used in their home, the authors should acknowledge this as a major limitation, and this underlines the possible use of a FFQ that was too difficult for the subjects. Authors should address this (maybe disclosing the percentage of "do not know" answers?)

3. Authors should describe the age range of the subjects for a better understanding in countries with different school systems

Author Response

Dear Reviewer,

Please see the attachment for my reply. Thanks very much!

Best regards,

Cleandy Lei

Reviewer 2 Report

Thank you very much for your interesting manuscript.

Line 46, it is better to provide statistics of brain damage caused by iodine deficiency in the world and in your country.

Line 58: How was the sample size obtained?

Line 60 Explain the method of random selection. Was it a lottery or did you have another method?

Line 78. Was this questionnaire made in advance? So, provide it with a reference. If you designed it yourself, do you mention that it was made by a researcher?

Has the questionnaire been validated? Or in this study? explain

Author Response

(The authors gave the same response as above.)

Reviewer 3 Report

1.      Is there any validation of the I-FFQ?

2.      How is the iodine calculated from the I-FFQ?

3.      Table 3. It would be informative to provide Q1 and Q3 for the median intake.

4.      Table 5. Please specify which gender is treated as the reference.

5.      How much of the daily iodine intake is contributed by the iodized salt? A following question is, in Table 6, males reported more iodized salt usage than female participants. Then the no-difference between genders in Table 5 would be interpreted with caution. Also need to be noted, the proportion of “unknown” are pretty high, ~80%.

6.      Table 6. The absolute Chi-square values do not provide useful information. Instead, p-values could be provided.

7.      When asking the iodized salt usage at home, a closely related question would be how often does the participant have meals at home. 

Author Response

(The authors gave the same response as above.)

Reviewer 4 Report

There are many spelling and grammatical errors in the essay.

Title:

Since no economic variable, even the occupation of parents, has not been measured and the study is limited to the relationship between socioeconomic variables, using iodized salt and iodine intake, it is suggested to modify the title.

Abstract:

 The following sentence is repeated twice in the abstract:

“They did not use iodized salt and consume iodine rich foods as main sources of iodine.”

 The following sentence does not have verb:

“Using iodized salt to cook food and appropriate intake of iodine-rich foods to prevent iodine deficiency disorders.”

Introduction:

The sentence starts with a number:

“80% of the iodine in the human body needs to be ingested from daily diet, 10-20% 34 from drinking water and less than 5% comes from the air”.

It is better to provide explanations about the Iodine deficiency in Macao and why a deficiency can cause complications in the introduction or method.

Methods:

Determining the sample size and sampling needs more explanation. How were the three schools chosen?

What was the age range of the samples?

How was the validity of the FFQ to determine iodine intake determined?

Regression analysis is recommended instead of correlation by entering all related variables.

Results:

It is better to present the consumption amount of each food group in Tables 3 or 4.

Discussion:

USI must be written in full on the first use.

Author Response

(The authors gave the same response as above.)

Round 2

Reviewer 1 Report

Authors responded to the main concerns in the original paper. Some of the new corrections need English editing.

Author Response

Dear Reviewer,

Thanks again for your comment. Please see the attached revised version of manuscript and supplementary file for consumption amount of each food item. Also, the manuscript has been already checked by a colleague fluent in English writing. I would be grateful if you could understand and satisfy my amendment. Please feel free to let me know if need further amendment and supplementary information. Thanks very much!

Best regards,

Cleandy Lei

Reviewer 3 Report

The authors have addressed my comments and concerns.

Author Response

(The authors gave the same response as above.)

Reviewer 4 Report

Methods:

Regression analysis is recommended instead of correlation by entering all related variables.

Results:

It is better to present the consumption amount of each food group in Tables 3 or 4.

Author Response

(The authors gave the same response as above.)
